# The experience of maternal mental distress in The Gambia: A qualitative study identifying idioms of distress, perceptions of contributing factors and the supporting role of existing cultural practices

**Katie Rose M. Sanfilippo**[1¤]*, **Bonnie McConnell**[2☉], **Buba Darboe**[3], **Hajara B. Huma**[3,4], **Vivette Glover**[5], **Lauren Stewart**[6]

1 Psychology Department, Goldsmiths, University of London, London, United Kingdom, 2 School of Music, The Australian National University, Canberra, Australia, 3 The Ministry of Health and Social Welfare, Banjul, The Gambia, 4 The National Centre for Arts and Culture, Banjul, The Gambia, 5 Institute of Reproductive and Developmental Biology, Imperial College London, London, United Kingdom, 6 Psychology Department, Goldsmiths, University of London, London, United Kingdom

☉ These authors contributed equally to this work.
¤ Current address: Centre for Healthcare Innovation Research, City, University of London, London, United Kingdom
* katie-rose.sanfilippo@city.ac.uk

## Abstract

Maternal mental health problems are experienced frequently in the perinatal period and can be up to twice as common in women from low- and middle-income countries. The terms used to describe mental illness symptoms and the factors that contribute to developing these symptoms are affected by culture and context. Based on 14 focus group discussions held with pregnant women, health professionals and cultural leaders, this qualitative study aimed to understand women's experiences of mental distress during the perinatal period in The Gambia. To do this it aimed to 1) identify the most commonly used idioms of distress, 2) identify the factors believed to affect women's perinatal mental health and 3) explore the role of existing cultural practices, including musical practices, that were identified as locally significant in supporting maternal and mental health. *Sondomoo tenkung baliyaa* (Mandinka) and *xel bu dalut* (Wolof) were identified as the most commonly used idioms of distress which roughly translate to lack of a steady mind/heart. Using thematic analysis, six themes (Poverty of the healthcare system, Shifting cultural context, Economic factors, Social factors, Spiritual factors, and Cultural practices involving music) were identified to describe the factors that shape women's experience of mental health during the perinatal period. Lack of economic resources, the prevailing poverty of the health system, an unsupportive husband and spiritual attack by evil spirits or witches were common reasons given for a woman experiencing maternal mental distress. Various existing cultural practices involving music, such as fertility societies, naming ceremonies and community music-making, were recognised as valuable for supporting women's mental health during the perinatal period. This work emphasises that initiatives to support perinatal mental health should be grounded in an understanding of local community cultural practices, knowledge, and experiences.

**Data Availability Statement:** The focus group discussion transcripts hold personal information and stories and therefore cannot be shared publicly. De-identified data that correspond to the results reported in this article may be made available only upon reasonable request from the National Centre for Arts and Culture, where the data is held. To request the data you can contact NCAC Director Hassoum Ceesay (hceesay@ncac.gm).

**Funding:** This study was funded by the MRC-AHRC Global Public Health: Partnership Awards scheme (MR/R024618/1 to LS). The writing of this manuscript was supported by a South East Network for Social Sciences/The Economic and Social Research Council funded postdoctoral fellowship (ES/V010158/1 to KRMS).The funders had no role in study design, data collection and analysis, decision to publish, or preparation of the manuscript. KRMS and BM's time and travel expenses were covered as a part of this project's funding. BD was paid for his time on the project. LS had her time bought out and HBH was paid as a full-time research assistant on this project.

**Competing interests:** The authors have declared that no competing interests exist.

## Introduction

Within the perinatal period, the time during pregnancy until a year after birth, maternal mental health problems and their symptoms are experienced frequently and, relative to high-income countries (HICs), can be up to twice as common in women from low- and middle-income countries (LMICs) [1, 2]. Various studies across sub-Saharan Africa have been conducted to understand local perceptions of perinatal mental health as well as risk factors and barriers to treatment [3–5]. Symptoms of common mental disorders (CMDs) such as anxiety and depression not only affect the pregnant woman but can also impact her developing infant and their future relationship [6, 7]. Transcultural psychiatry acknowledges that mental illness symptoms are expressed differently depending on culture and setting [8]. For instance, perinatal CMD symptoms are usually described behaviourally rather than cognitively in sub-Saharan African contexts and the terms used are generally more somatic [9, 10]. For example, in a South African context, 'idioms of distress' frequently involve understandings of mental distress as being caused by factors external to the individual [11]. These cultural perspectives on mental distress shape the way individuals experience challenges during the perinatal period, as well as what coping and support strategies might be most helpful.

Research has shown that factors that contribute to developing CMD symptoms are affected by culture and context [8]. Largely, the risk factors for developing perinatal CMD symptoms are biological, experiential, psychological, social, and cultural [12]. However, there are significant differences in the risk factors found in LMICs compared with HICs [1, 2, 13]. Poverty and economic stress, experienced more by women in LMICs, can increase a woman's risk of developing high levels of perinatal CMD symptoms [1, 14, 15]. In the sub-Saharan African context, a study by Wittkowski et al. [13] identified four main factors contributing to postnatal depression symptoms: lack of social support (especially from the husband), relationship problems, an unwanted pregnancy, and cultural factors, such as family structures or polygamy.

A study by Sawyer et al. [16], based in Old Jeshwang in The Gambia, explored women's experiences during the perinatal period. Women felt that the perinatal period represented a transition to adulthood. They discussed the various physical difficulties and the serious physical threat of giving birth, the importance of having a child, especially a male one, the financial strain of having children, and the feeling of being 'on their own' during the perinatal period. Unlike the current study, Sawyer et al. [16] only recruited from one region of The Gambia and did not specifically focus on women's experiences of perinatal mental distress or their perspectives of contributing factors.

Like many countries around the world, there is rapid social and cultural change happening in The Gambia, which has affected women's experiences and the challenges faced during the perinatal period. For example, women now increasingly partake in supplementary income-generating activities such as vegetable gardening or selling goods at the market [17, 18]. While for some women these changes have resulted in increased economic independence and wellbeing, they have also generated new challenges and tensions within the family as the gendered division of labour is renegotiated [19].

Studies have also demonstrated that social and cultural factors can be protective against developing perinatal CMD symptoms in LMICs. Fisher et al. [1] suggested that higher levels of education, a permanent job, being in the ethnic majority and having a kind and trustworthy partner were protective factors [1]. Strong social support networks have been found to mediate maternal mental health outcomes [20], acting as another potential protective factor. Group music making has been identified as an untapped public health resource, with studies showing that engaging the community in group music making has beneficial effects for the individual and the wider society [17, 21, 22] through its ability to create social bonds, increase mood and

reduce anxiety and depression symptoms in the general population [23] and for those in the perinatal period [24].

The Gambia is a medically pluralistic society. People use a variety of health care options, including biomedicine (by attending their local clinics), indigenous herbal medicines, and spiritual treatments [17]. The World Bank estimates a $31 per capita health expenditure per annum, one of the lowest in the world [25] and many women experience a lack of access to healthcare, highlighting a need for solutions that are culturally embedded, low-cost, and sustainable. Traditional cultural practices in The Gambia are highly diverse, including a range of musical practices that are specifically oriented toward communication, social support, and healing [26, 27] These cultural practices are led by specialists such as griots, defined as hereditary musical experts and knowledge holders who play important cultural roles as communicators, emotional transformers and conflict mediators [17], and Kanyeleng fertility societies that are usually comprised of women who are dealing with or have dealt with infertility or child mortality and who perform music as a central part of their activities [17, 28]. As part the World Bank-funded 'Women in Development Project', the Ministry of Health and Social Welfare commissioned groups of traditional communicators (TCs), built on pre-existing Kanyeleng groups, to help disseminate health related information [29]. The reproductive health expertise, musical performances, and health communication practices of the Kanyeleng/TC groups represent a nexus whereby biomedical and traditional models of perinatal health care meet [30]. Despite their cultural importance for women during the perinatal period, this study is the first to examine Kanyeleng societies and related cultural practices in relation to maternal mental health.

This study aimed to understand Gambian women's experiences of maternal mental distress by identifying the most commonly used idioms of distress and the factors believed to contribute, either positively or negatively, to women's perinatal mental health. It also examined local knowledge systems and cultural practices involving music in relation to maternal health and mental health that have been neglected in existing research, despite their cultural significance for women in the Gambia. This research was able to meet the aims of this study.

## Materials and methods

A qualitative approach was undertaken to understand women's experiences of perinatal mental distress including potential contributing factors, relevant idioms of distress, and the role of embedded cultural practices in maternal and mental health. Focus group discussions (FGDs) were the primary mode of data collection. The Consolidated Criteria for reporting qualitative research (COREQ) [31] was used and the COREQ checklist can be found in S1 Checklist.

### Study team

The project emphasised the development of strong collaborative partnerships that involved critical reflection on the positionality of members of the research team and differences in disciplinary orientation and culture. The study team is international and interdisciplinary. It includes experts from the Ministry of Health and Social Welfare (MHSW) (Buba Darboe (BD) and the Centre for Arts and Culture (NCAC) (Hassoum Ceesay (HC)) in the Gambia. It also consists of researchers from the UK and Australia specialising in music and health (Katie Rose M. Sanfilippo KRMS), music cognition (Lauren Stewart (LS)), perinatal psychology and biology (Vivette Glover (VG)), and Gambian music and culture (Bonnie McConnell (BM)).

BM (female), an ethnomusicologist, and BD (male), head of health communication at the Ministry of Health, have extensive expertise running focus groups in The Gambia. BM trained the research assistants (RAs) (Hajara B Huma (HBH) (female) and Malick Gaye (MG) (male))

in running semi-structured FGDs. Both HBH and MG are trained psychiatric nurses working in The Gambia. At least two members of the research team who spoke Mandinka and/or Wolof (BM, BD, HBH, MG) were present at each FGD. BD, BM both have extensive experience working with Kanyeleng groups and therefore had some professional relationships with the participants. KRMS (female) spent time working with colleagues in The Gambia but did not run the FGDs as she does not speak either local language.

## Methodological approach

The research team (KRMS, BM, BD, HBH, MG) engaged in participant observation (2018–2019) to gain an understanding of tacit knowledge relating to cultural ceremonies during the perinatal period. Participant observation involved regular attendance and participation in naming ceremonies (*kunliyoolu*, Mandinka) and Kanyeleng performances of various kinds, such as initiation ceremonies for new Kanyeleng (*kuroo*), and special Kanyeleng naming ceremonies for children born to Kanyeleng women. While field notes were not included in the analysis, participant observation provided the necessary contextual knowledge through which to understand the examples and concepts discussed in the FGDs.

This research team ran FGDs with various key stakeholders in perinatal and mental health, including pregnant women, community birth companions (CBCs), midwives, griots, and Kanyeleng/TC groups. A total of 14 FGDs (n = 114 participants in total) were conducted. Focus group participants were chosen for their ability to provide insight into the nature of mental distress in the perinatal period.

The methodological approach was designed to engage marginalised perspectives that have not been captured in existing research on women's mental health in West Africa, despite their local cultural significance and influence. It employed a collaborative approach, through our partnership with the MHSW and the NCAC, that aimed to enable the research to be enriched by interdisciplinary research expertise while remaining grounded in the priorities of Gambian partner organisations and research participants. The interdisciplinary approach aimed to take local knowledge systems, held by local women's groups like the Kanyeleng, as the starting point for building an understanding of women's experiences during the perinatal period. Participant groups were selected to incorporate cultural expertise as well as knowledge of women's health in the local context.

## Ethics

Ethical approval was obtained from the Goldsmiths University Ethics Committee, The Gambia Government/MRC Gambia joint Ethics Committee and the Australian National University Ethics Committee. All participants were given information about the project and provided oral informed consent.

The nature of the research involves the potential of sharing personal information surrounding mental health and pregnancy. It was possible that some of the themes involved in the questionnaires or FGDs could lead participants to reveal episodes or thoughts of self-harm or intimate partner violence. If this was the case, the woman received immediate counselling with the RAs, both of whom are trained psychiatric nurses. If needed, women who revealed episodes or thoughts of self-harm were referred on to the Community Mental Health Team (CMHT) for further management. If the CMHT deemed it appropriate, they could refer the woman on to the psychiatric team.

## Setting and participants

The current research included both rural and urban areas in the Western and North Bank regions. Research sites were defined based on geographic location and were selected, through

**Table 1. FGD group type, represented area, number of participants and language.**

| Group Type | Area | n | Language |
|---|---|---|---|
| CBCs | Kerr Omar Saine | 8 | Wolof |
| CBCs | Kissimajaw | 8 | Wolof |
| CBCs | Farato | 6 | Mandinka |
| Kanyeleng/TC | Sanyang | 10 | Mandinka |
| Kanyeleng/TC | Ndungu Kebbeh | 10 | Wolof |
| Kanyeleng/TC | Jambangjelly | 10 | Mandinka |
| Kanyeleng/TC | Fass Njaga | 4 | Wolof |
| Midwives | Brikama/Gunjur/Kafuta | 5 | English |
| Midwives | Serekunda/Sukuta/Fajikunda | 6 | English |
| Musicians/Griots | Serekunda | 10 | Wolof |
| Musicians/Griots | Serekunda | 10 | Mandinka |
| Pregnant Women | Sinchu Baliya | 8 | Wolof |
| Pregnant Women | Brikama | 10 | Mandinka |
| Pregnant Women | Fajikunda | 8 | Wolof |

$n$ = total number of participants present at the FGD. Group affiliations are not exclusive. For example, some participants within the CBC or musician/griot FGDs are also members of Kanyeleng or TC groups. No additional demographic characteristics were collected. However, a previous study conducted by the same research team collected more detailed demographic data about the pregnant women attending the same antenatal clinics. More information about the pregnant women who attend these clinics can be found in Sanfilippo et al., [32].

consultation with our Ministry and NCAC partners, to include participants from two major language groups, Mandinka and Wolof. A convenience sample of 114 participants (Table 1) were involved in the FGDs.

Community birth companions (CBCs) were chosen because they have extensive knowledge and experience of the local health system and how traditional and biomedical treatments are used simultaneously. Midwives were chosen to represent the trained and practising relevant clinicians within the government health system. Griots were chosen for their insight into cultural practices and the role they play in health. Kanyeleng/TC groups were chosen as experts in music associated with health communication and reproductive health. Kanyeleng women have frequently had a negative experience during pregnancy, birth, or postpartum, so they could also discuss their personal challenges. Pregnant women were chosen to represent current experiences of being pregnant.

CBCs, midwives, and Kanyeleng/TC groups were all invited by phone and in person to participate by our MHSW collaborator (Buba Darboe). Musicians and griots were invited by phone and in person to participate by our NCAC collaborator (Hassoum Ceesay). Pregnant women were invited to participate in person by nurses and midwives during their reproductive and child health clinic visits across three antenatal clinics (Table 1).

## Data collection

All FGDs were conducted between May and September 2018. Each FGD lasted approximately an hour and was led by members of the research team fluent in Mandinka or Wolof (BD, HH, BM). Only research team members and participants were present. The questions followed a semi-structured format. Topics included challenges encountered in pregnancy, experiences of perinatal distress, the support offered or sought out by women facing difficulties during pregnancy, existing cultural practices associated with the perinatal period, and the role of cultural

musical practices in supporting health generally and women's perinatal mental health. The questions were adapted for the different discussions depending on the expertise of the informant group (See S1 Text for the FGD guides). Each FGD's audio recording was transcribed in the original language and translated into English by experts from the NCAC.

## Analysis

A reflexive thematic analysis [33] was performed to identify categories and themes that could be used to represent the data. The flexibility of this analysis approach allows for a synthesis of the data concerning a specific research question. Following Braun and Clarke's steps [33], after transcription, specific patterns within the data were identified and written down as a set of codes in English with key terms kept in Mandinka or Wolof. This was completed by the two research assistants (RAs), both psychiatric nurses from The Gambia and fluent in Mandinka and Wolof. Then the English translations of all the FGDs were reviewed by KRMS. After this initial review, patterns and broader categories within the data were identified to create a set of initial codes. These codes were then merged with the codes identified by the RAs to ensure the local terms and expressions were not lost. A second review of the codes and categories was undertaken by BM where all the identified codes and any incongruities were discussed as a team to ensure all significant codes and categories were represented. Once the codes were finalised, they were synthesised into broader themes, which were then defined and named. The analysis was completed within Dedoose [34].

## Results and discussion

The overall aim was to understand women's experiences of mental distress during the perinatal period in The Gambia. Within this section we will first discuss the various idioms of distress identified. We will then present the themes that help to identify the factors believed to affect women's perinatal mental health and explore the role of existing cultural practices involving music that were recognised as locally significant in supporting maternal and mental health.

   Part of our analytical approach was to ensure that local language terms were noted throughout the analysis procedure. This was to help inform the analysis of this work as well as to inform the development of a translated tool used to measure symptoms of maternal distress in The Gambia [32]. Conceptualisations of distress and stigma were highly complex and varied between individuals and between different specialist groups. A constellation of terms and phrases (idioms of distress) were identified to describe some of the emotional states a woman can experience in pregnancy and after birth. Our understanding of these idioms of distress is based on the terms used across all the focus group discussions conducted in the Mandinka and Wolof languages (the midwives FGDs were conducted in English and therefore did not inform our understanding of local language idioms of distress). These terms highlight the tendency to attribute mental distress to factors external to the individual. For example, *niikuyaa*, the term for sadness in the Mandinka language, is frequently used to describe a sad event experienced by an individual, rather than an internal emotional state. Other terms frequently mentioned referred to feelings of nervousness (*kijafaroo*, Mandinka; *tiit*, Wolof), and worry or "too much thinking" (*miraalisiyaa*, Mandinka; *xalat bu bari*, Wolof) (for more detail around the different teams identified see S2 Text). These terms were found to be encompassed by two umbrella terms, *sondomoo tenkung baliyaa* (Mandinka) and *xel bu dalut* (Wolof). *Sondomoo tenkung baliyaa* translates to "lack of a steady/calm mind/heart" and *xel bu dalut* translates to "lack of a peaceful mind". These were found to be the most helpful and non-stigmatising idioms of distress to use when talking about CMD and their symptoms in The Gambia. Two common terms used to describe mental illness (*sondomoo kuurango*, Mandinka; *febar xel*, Wolof) were

found to have more stigma attached and were therefore less appropriate for discussing experiences of perinatal mental distress.

We also aimed to understand the factors that contribute to women's positive or negative experiences of mental health within the perinatal period. The externalisation of mental distress was also evident in the discussion of these factors. The main themes that were identified include: (1) Poverty of the healthcare system (2) Shifting cultural context, (3) Economic factors, (4) Social factors, and (5) Spiritual factors. The sixth theme, (6) Cultural practices involving music, specifically addresses our aim to explore the role of existing cultural practices, especially those that involve music, in supporting maternal and mental health.

Themes 1 and 2, 'Poverty of the healthcare system' and 'Shifting cultural context', describe the broader conditions that shape women's experiences, and therefore mental health, during the perinatal period. Themes 3–5, 'Economic factors', 'Social factors', and 'Spiritual factors', were found as intersecting, influencing each other and subsequently the individual women's mental health. Theme 6, 'Cultural practices involving music', encompasses the way women draw on these types of cultural activities to manage social and spiritual challenges, and to counter the negative effects of cultural change by providing a sense of continuity and meaning. This theme was also found to interact and intersect with all themes, highlighting the way cultural practices are woven throughout women's daily lives and experiences during the perinatal period.

## Poverty of the healthcare system

This theme represents aspects of the broader setting that were frequently discussed in the FGDs. Specifically, participants expressed concerns about the poverty of the healthcare system and its lack of resources. This theme highlights the strong influence of macro-level structures in shaping women's experience of the perinatal period, health, and mental health [35].

Approximately 50% of women in The Gambia fall below the poverty line [17, 36]. However, the poverty of the country's health system and the lack of accessibility to health resources affect all women in The Gambia. The midwives discussed at length how a lack of resources was affecting the clinics and consequently women's perinatal care. Midwives explained how the overcrowding of the clinics led them to rush their appointments with women, not giving them enough time to discuss some of the women's needs beyond their immediate physical ones.

> *"[A pregnant woman] will have stress at home and if they come maybe you will not have that time to discuss for a lengthen period with the midwife. Because the midwife don't have that time, you have over a hundred patients waiting for you and they are saying be quick I am in a hurry. The woman [. . .] at the end of the day, she will develop stress and will come to [have poor] mental health."–Midwife from Serekunda FGD*

Midwives also discussed the poor structure and organisation of the clinics themselves. They described the clinics as understaffed and under-resourced, leading to long wait times in usually very hot and ill-ventilated rooms.

Rates of maternal and infant death are still relatively high. A total of 433 out of 100,000 live births resulted in maternal death and 63 out of 1,000 births resulted in infant death in the first five years of life in 2018 [37]. This reality was a salient worry discussed by many FGD participants as impacting their mental health and wellbeing.

> *"Your mind is always focused on whether or not you will die. When you give birth, you live but your child doesn't live, that causes a person to suffer. You can give birth as well, your child lives but you die. Those are all difficulties of pregnancy."–Mandinka Kanyeleng from Sanyang*

Overall, poverty of the healthcare system sits as a wider permeating condition impacting the health care system, which was described as not well equipped to treat women's health and mental health effectively.

## Shifting cultural context

Another important contextual factor that was identified in the FGDs was the effect of culture, its norms and expectations and the way it is shifting due to urbanisation and westernisation. Culture, traditional gender roles and family relationships were described by participants as important for maintaining social stability and individual wellbeing. Acting against cultural norms was seen to be detrimental to women's mental health. As explained in several FGDs, if a woman were to act outside of these expectations, like giving birth out of wedlock, this could have a negative effect on her mental health, especially as it may result in a loss of her family's support.

Some participants explained that women are traditionally viewed as incomplete and in need of support throughout their whole life, not only from her family but also from others within the community. One pregnant woman from Sinchu Baliya explained how this belief dictated why women need to move from their parents' home to their husband's home when they get married. It also gives some insight into power relationships between a woman, her in-laws and husband.

*"You know if women were complete, they would be in their parents' homes. No man vacates his house to go to a woman's house in order to get married. That is, therefore, an indication that women are incomplete. You men are stationed at your homes waiting for us so you can complete us. So if you complete our weakness and give us support, it will be good. But imagine being pregnant coupled with stress and all kinds of sorrow; if that persists for long, it can even bring you some other problems as well"–Wolof pregnant woman from Sinchu Baliya*

Other cultural norms and practices were discussed as impacting a women's mental state during pregnancy and after birth. Having a son was described as similar to having a 'pension', as it is a male child's responsibility to care for his parents in their old age. FGD participants explained that if a woman has not conceived and given birth to a boy, it is usually the woman who is blamed, and this can contribute to mental distress. Additionally, the practice of child marriage was also described as having a detrimental effect on girls' and women's mental health.

It was also revealed how westernisation and urbanisation have shifted women's roles and values. The high level of rural-to-urban migration has shifted many families away from an agricultural livelihood [38]. This migration from rural to urban areas has also influenced a perceived increase in Western values. Some participants, mostly from the informant groups embedded in traditional beliefs and practices (e.g., CBCs and griots), talked about how Western values have influenced the desires and values of the younger generation, with more women wanting to be like *toubabs* (outsiders or Westerners). Participants described these shifts in values to include matters such as what clothing people wear, how they interact with their husbands, what they eat as well as what education they want for themselves and their children. Some viewed these changes as positive, leading to increased freedom and opportunities for women. Other participants viewed them as causing more financial and mental strain on mothers as gender roles and behaviours are renegotiated, potentially leading to conflict within the family and increased responsibilities for women both within and outside the home.

*"The present generation is different from the past generation. In the past generation people lived on their sweat, but now people live off a bag of rice. When it finishes you buy another*

*one. Now you are thinking of feeding the family, clothing, and education. Now on the side of education things were not like as it is today. In those days education was not compulsory, people just attended the local Qur'anic school just to learn how to pray. Now everybody wants to be toubabs (Westerners) [and] that brings problems."–Mandinka CBC from Farato*

Overall, shifts in cultural values, traditions, and beliefs are embedded contextual realities discussed as significant factors influencing women's experiences during the perinatal period, interacting with all factors describe below. They shape women's access to care, what care they receive, as well as their roles, expectations, behaviours, and relationships.

## Social factors

Social support from the immediate family, as well as from the wider community, was discussed as an important factor that affects a woman's mental health during pregnancy and after birth. A challenging relationship or conflict within the family was explained as a common cause of mental distress. Within the family unit, a woman's relationship with her husband was consistently described as one of the most important factors affecting her mental health.

*"I am pregnant at the moment, but since the beginning I always have conflict with my husband. That sadness [i.e. connected to conflict with husband] has bothered me a lot. [. . .] When I am pregnant I suffer from sadness, I don't have peace of mind."–Mandinka pregnant woman from Brikama*

While the relationship with the husband was discussed as the most influential, other relationships in the home were also addressed. Many women live with one or more co-wife and their in-laws. Polygamy was regarded as an accepted practice for many women. However, the FGDs highlighted strong concerns about lack of control over family decisions and unequal treatment of co-wives.

The uneven distribution of not only social support, but also resources, was explained to cause some women mental distress during the perinatal period.

*"Men are not honest. You marry a second wife and place her above your first wife. [. . .] When [he is] in the room with one of them [he tells] her sweet words or convincing words. If God blessed you with pregnancy during that period of time, you will be miserable. You will be left without being able to eat well. That's when illness takes you"–Mandinka Kanyeleng from Jambanjelly*

The stress that can be caused by conflict within the family was explained to produce a potentially harmful effect on the unborn baby.

*"[Conflict with] your husband, co-wife or even your family. You know that causes stress to an individual as well as her unborn baby. And that is not good in pregnancy."–Wolof pregnant woman from Sinchu Baliya*

Conflicts within the family emerge in the context of a gendered division of labour and women's extensive domestic responsibilities. Participants discussed how unequal power relations within the family mean that individual women may not have the freedom to make their own care decisions. Therefore, social relationships were also described as factors affecting a woman's access to care, impacting their mental health and potential mental health care. Some women depend on support and permission from others in the family (e.g., husband or in-

laws) to receive antenatal care. Many midwives talked about the importance of involving husbands in care decisions.

*"When the woman talks to the husband alone, he will not take it seriously he will say "[it] is expensive I cannot do that". [. . .] Sometimes we do call the husband and we talk to them and they say "yes I can do it." [. . .] So the support we are giving right now is health education, inviting husbands to come so that we can talk to them at an earlier stage so that we will not end up with a problem. I think this too is very important."–Midwife from Serekunda FGD*

Relationships with family, friends or others in the community were also discussed as helping to create supportive networks. Many participants talked about the importance of having someone to talk to from the community, expressing concerns about increasing social isolation contributing to women's mental distress during the perinatal period.

*"Now if you are in that condition and you decided to seclude [yourself] in your house, no one gives you help [. . .], comforts you so that you can regain your health, [tells] you [to] eat well and help yourself and your unborn child. If you don't have that, it would become a big problem. Yes, that is among the first problem of the pregnant woman. And nowadays such a situation is too much."–Wolof TC from Ndungu Kebbeh*

The dynamic elements of the family structure, involving shared living with co-wives and in-laws, creates an environment that can both support and hinder a woman's mental health. These social factors were described as affecting not only a woman's mental state and access to care, but also her economic situation and spiritual wellbeing.

## Economic factors

Participants described an individual's access to resources as influencing her mental state during the perinatal period. Within this paper, we distinguish two forms of poverty: poverty of the health care system (theme 1) with implications for the level of health provision, and poverty of the individual, or a lack of resources impacting a woman's ability travel to clinics and the inability to obtain food and medicine. The differentiation between the poverty of the healthcare system and the poverty of the individual was evident within the FGDs. Economic factors were usually one of the first problems discussed. For example, a midwife from Serekunda talked about how she had personally tried to help a woman who did not have the money to travel to the clinic.

*"If they come to the facility you tell them to buy [a] pad and they don't have fare to come, sometimes is very stressful[. . .] if you have something you can give them something to use that as the fare. But how many people can you do it for?"–Midwife from Serekunda FGD*

Many women depend on their husband's income and must ask permission to use any funds. Not feeling in control of one's income and finances was discussed as negatively impacting women's mental health and their relationships with their husbands. This lack of control was also shown to affect the social occasions women value. Held one week after the infant is born, the naming ceremony is a highly significant social occasion for a new mother. Naming ceremonies typically incorporate music and dancing, featuring griots, Kanyeleng, and other performers, and it is a proud and joyful moment for many mothers. Along with music and food, having new clothes to wear during a naming ceremony is part of the performative celebration of a new mother, reflecting the broader social significance of clothes in notions of

femininity and women's social networks in the Gambia. While it can be a moment of joy, participants also talked about the stress and sadness that women can experience when they are not able to afford a suitable naming ceremony.

> *"When she gets a poor husband with low income, she would be troubled by the fact that he may not be able to perform a befitting naming ceremony for the child. [. . .] As long as she is not certain of what the husband is going to do, she will not be at ease. That can cause too much thinking in her for she needs new clothes yet she can't afford it. That alone can cause stress in a woman."–Wolof CBC from Kerr Omer Saine*

In these examples, social factors (e.g. the relationship with the husband) interact with economic factors and cultural expectations. While these examples show the negative effect of this intersection, money-sharing societies were also described as having a positive effect. These involve women from the community gathering together to support each other financially. Each woman contributes a little at each meeting. On a rotating basis the sum of all the contributions goes to one member of that group. These societies were described as creating a network of social and financial support for women in general and especially during the perinatal period. Furthermore, other social gatherings, such as the previously mentioned naming ceremonies, were also explained as a way to support women and their families financially. All these types of gatherings include many different types of participatory music making, showing the integration and importance of music in these cultural and social celebrations.

Social and economic factors interact in ways that can be both helpful and detrimental to a woman's mental health. A woman's financial situation is frequently contingent on her husband's income, leaving many women feeling financially dependent and without spending agency. However, various social practices involve the reciprocal sharing of resources that create supportive social and financial networks for women during the perinatal period. These practices of social support, including the important naming ceremony, are often described in relation to Islamic ideas of care and generosity.

## Spiritual factors

Spiritual factors were believed to explain mental distress symptoms, such as irritability, as well as to be the cause. A woman's faith in God was described as a means to accept and cope with the difficulties she may face during the perinatal period. Religion and spirits were usually discussed together. For example, when a spirit afflicts someone, prayer, along with other traditional practices such as music making, were described as one method used to help chase a spirit away.

In a context where over 90 percent of the population is Muslim, references to Islam were woven throughout the FGDs, framing women's experiences of mental distress in relation to broader ideas about faith and morality. At the same time, the discussions reflected syncretic, localised understandings of spirituality, drawing on traditional beliefs as well as Islam and Christianity.

Many people emphasised the importance of faith in God, indicating a belief that their fate is in his hands.

> *"You begin to ask certain question such as, what shall we do when I delivered? You know all that is a divine destiny but due to your anxiousness you begin to say this. You are just being optimistic in your desires. However, the ultimate decision rest with God for final determination of his decree as no one knows the future."–Wolof musician/griot*

Participants also talked about using their faith as a form of acceptance rather than fatalism.

*"When God caused you to finish your childbearing altogether and willed that you not bear a child, you will not have any."–Mandinka Kanyeleng from Jambanjelly*

The majority of people in The Gambia believe in spirits and witches [39, 40]. Spirits and witches are believed to either guide, haunt, or curse [17, 40]. Along with *marabouts*, Muslim religious leaders, and other cultural leaders such as village elders, Kanyeleng groups perform rituals which involve music making or use traditional medicines that aim to drive away evil spirits [17, 40]. Participants discussed being fearful of spirits (*jinns*) or of having a spiritual attack. They explained that being attacked by an evil spirit could change a woman's behaviour by making her angry or irritable and cause her mental distress.

*"[If I am being attacked by a spirit], even if you just speak to me, I will get angry [. . .] [An evil spirit] appears to some during the labour period, and if God does not help you, the child can die. It appears to some people when they are in a crowd. [. . .] It (the spirit) can make them fall down, and sometimes they can have a very serious injury. [. . .] She can be aware that 'today an evil spirit knocked me down'. But during the time it was inside her, even if you cut her arm, she would not be aware."–Mandinka musician/griot*

Some participants expressed how when symptoms were attributed to spiritual attacks, this could cause delays in accessing appropriate care for diseases such as malaria.

*"Because malaria can cause hallucination, people would then begin to associate it with spirituality. People would speculate by associating it with every kind of myth possible when it is nothing but malaria. As a result, instead of taking the victim to the doctor, she is instead taken to the witch doctor."–Wolof CBC from Kissimajaw*

The spirit husband (*kuntofengo*) was described by many participants, especially griots and Kanyeleng whose traditional practices commonly involve fending off this spirit. It was explained that it is during pregnancy and labour when the spirit husband usually disturbs women. Spirit husbands are believed to cause infertility, miscarriage, and infant mortality. The fear of the spirit husband, or believing that he has already affected previous children, caused distress for some participants during their pregnancy. One Kanyeleng woman talked about how the spirit husband caused her to experience both psychological and physical symptoms.

*"The spirit husband can cause a lack of peace of mind. I involved myself in the Kanyeleng because of the spirit husband. My children are burned by the evil spirit. I sometimes dream of having sexual affairs with him. When I am pregnant and I have that dream, that alone causes my stomach to hurt. When this happens, I panic. I still have that problem."–Mandinka Kanyeleng from Sanyang*

Overall, the role of spirituality was discussed at length by all informant groups, showing its important role in women's mental health during the perinatal period.

While some contributing factors to a women's mental state during the perinatal period were attributed to the individual such as physical health, previous experience, and education, most participants talked about factors that are external to the individual. Perinatal mental distress was most frequently attributed to social, economic, and spiritual factors that may be out of an individual woman's control. These various factors are understood to interact and

influence a woman' mental health during the perinatal period in the context of broader societal experiences of poverty of the healthcare system and a shifting cultural context.

## Cultural practices involving music

Across all the FGDs with the different informant groups, cultural practices involving music were described as important activities that take place during the perinatal period and intersect with the contributing factors discussed. In particular, musical practices were described as effective in maintaining a sense of cultural continuity and social connection in the face of challenges such as conflict, spiritual attacks, and changing cultural norms. Furthermore, cultural practices involving music were identified as integral to supporting health and wellbeing.

> *"You ask if music has the ability to help someone with a problem, [with being] depressed or a sick person? Yes, in fact that's the reason for the existence of music. In such a situation, where one is afflicted with problems, it will now be suppressed drastically. Whoever hears a song whilst in a state of depression shall enjoy a degree of relief."–Wolof musician/griot*

Within the FGDs with the Kanyeleng/TCs and griots it was explained how music is necessary for women during the perinatal period. Singing or listening to music allows women to feel energised while also relaxing their mind. It was explained how singing benefits the pregnant woman and has a direct influence on her growing foetus.

> *"It is said that songs are necessary for a pregnant woman. Songs are part of things that rejuvenate her. [. . .] Any music that she feels can entertain her and that shall relieve her or that can make her happy, such a song is good for her [. . .] She can be listening to it so as to achieve tranquillity in her state of mind [and] in her body as well as in her unborn child."–Wolof musician/griot*

Cultural musical practices were discussed as intersecting with the various spiritual and social factors discussed above. For example, social gatherings such as naming ceremonies were described as flavourless without the inclusion of music by griots.

> *"So if you have ceremonies and you do not see such griots or your relatives, then you must assess yourself once again. The absence of griots in our functions are like food (specifically benachin, a traditional rice dish) without salt. The food will be tasteless. That is why getting together softens the hearts. It causes happiness." Wolof TC from Ndungu Kebbeh*

These types of cultural gatherings were described as important in bringing people together, resolving conflict, and building social support. Additionally, engagement with musical practices was often described as closely connected to other aspects of culture, such as food. Kanyeleng ritual is a site where song, dance, food, comedy, and prayer are combined with the goal of promoting social and spiritual health.

The rituals and performances of the Kanyeleng were also discussed as important in supporting and giving advice to pregnant women and new mothers, specifically around their role in addressing the spiritual factors described above. Infertility and infant mortality are believed to frequently result from the presence of a spirit husband whose jealousy prevents a woman from having a child [40, 41]. To evade the *kuntofengo*, Kanyeleng groups participate in prayer, disguise, trickery, music performance, and rituals aimed to prevent infertility and infant deaths [28, 40].

In addition to these ritual practices that have social and spiritual significance, Kanyeleng groups also play an active role as health communicators where they use songs to share health

information with people in their communities. This was seen as important in the context of limited access to health information and care.

> *"Some songs [. . .] are advice songs for the mother. When you sing those advice songs for the pregnant women and new mothers, you will see [that] those that don't have understanding will get it. If [I think] my child should stop going to the nurse after three months, when Kanyeleng go to the naming ceremony and sing the song there, it will make that person aware that a child should go to the nurse until they are five years old. But if I didn't know that, I stopped at just three months, my child's vaccinations will be left behind."–Mandinka Kanyeleng from Sanyang*

Traditional songs used during the perinatal period, such as lullabies, were described in many FGDs as having a positive effect on the mother's and infant's well-being. Lullabies were explained to serve numerous functions. They can help the infant stay quiet or fall asleep so the mother can focus on other tasks or be used as a way for the mother and her child to bond with one another. While lullabies were discussed as helpful musical practices used during the perinatal period, many participants talked about them being increasingly forgotten by current generations. It was explained how mostly older women knew these types of songs, showing the way in which culture is shifting, impacting not only women's mental health but the musical tools they might use to connect with their children.

Overall, music is present in women's everyday lives and within important cultural practices during the perinatal period. Music engagement influences women's relationships with their community, through ceremonies, and with their infant, through lullabies. Music impacts women's mental health during pregnancy, through individual listening and singing or through the social support networks it fosters. Music's place in cultural practices was identified as a thread which runs through all the factors described as impacting a women's mental health during pregnancy and after birth. Music plays an integral role within money sharing societies to help with fundraising, at naming ceremonies to support the new family and the community, and is used to fend off evil spirits. Therefore, within The Gambia, cultural practices involving music were discussed as a way to address common factors believed to impact a woman's mental health during pregnancy and after birth.[NO_PRINTED_FORM].

## Conclusions

Overall, this study brings a new perspective to a neglected public health issue that centres on the voices of women and their existing cultural health practices that take place within the community. This current work adds to the evidence base surrounding the importance of local context and cultural practices around maternal health, mental health, and well-being. This qualitative study not only identified the idioms of distress relevant in The Gambia but was able to identify six themes (Poverty of the healthcare system, Shifting cultural context, Economic factors, Social factors, Spiritual factors, and Cultural practices involving music) to encompass the different elements which are believed to contribute to women's mental health during the perinatal period. Conducting FGDs with a variety of stakeholders, including pregnant women, health professionals, and cultural leaders across a variety of settings, allowed for different perspectives and experiences to be considered. While all the main themes were discussed across all informant groups, especially the role of social and economic factors, midwives discussed more the role of poverty in the health care system. Kanyeleng, griots and CBCs discussed more the role of cultural and musical practices and the impact of spiritual beliefs and Kanyeleng and pregnant women discussed more their individual experiences and the interaction of all these factors together.

Overall, two terms were found to best describe CMDs and their symptoms in a non-stigmatising way: *sondomoo tenkung baliyaa* ('lack of a steady/calm mind/heart'; Mandinka) and *xel bu dalut* ('lack of a peaceful heart'; Wolof). The way CMDs and contributing factors were discussed demonstrated an understanding of perinatal mental health as being dependent on social, economic, and spiritual factors external to the individual. The perceived negative and positive impact of these factors on women's perinatal mental health were presented together. This study highlighted the way these social, economic, and spiritual factors interact with one another and the permeating effect of the wider poverty of the health care system and a shifting cultural context.

Poverty was seen to affect the resources available to health centres and antenatal clinics, impacting women's access to care across the country. The midwives explained that appointments were usually rushed and the clinics were crowded and understaffed. Similar to studies in other African contexts (e.g., a study conducted in Uganda [4]) health system factors such as low staffing, rushed visits, and lack of trained professionals, community level factors such as lack of a referral system, and an individual's access to resources, such as ability to travel to the clinic, were discussed as barriers to perinatal care. Also, comparable to other previous research conducted in sub-Saharan Africa [2, 5, 14, 42], economic strain, lack of spending autonomy, and poverty were identified as significant contributing factors to perinatal mental distress. While poverty was discussed in some previous research as a contributing factor of perinatal CMD symptoms or mental distress, the current study differentiated two forms of poverty, highlighting the wider permeating experience of poverty that shapes all women's experiences during the perinatal period.

Previous qualitative studies have identified social factors that contribute to perinatal mental distress in LMICs, with specific mention of the importance of husbands [3–5, 43]. Whether it is an overall lack of support or a specific conflict with the husband, studies from Uganda, Ethiopia, Nigeria, South Africa, Ghana, Malawi and other LMICs have identified the relationship with the husband as one of the most important influencing factors contributing to poor perinatal mental health [3–5, 13, 14, 43–45]. Consistent with previous research in The Gambia [16, 46, 47], the importance of husband support was identified as one of the most salient factors in our study. This speaks to the need for perinatal mental health services and interventions in LMICs to engage fathers or at least help women navigate this important relationship.

In line with Wittkowski et al.'s [13] review, cultural values, beliefs, and expectations were found to impact women's perspectives on their mental state during pregnancy and after birth. Cultural factors such as family structures [13], polygamy [45, 48], the stigmatisation of being single [45] and the preference for a male child [3, 45] have also been identified as contributing factors to perinatal mental distress in other studies across sub-Saharan Africa. This study situated various cultural changes, such as women participating in additional work, and concerns about westernisation and shifting values, as wider factors influencing women's mental health during the perinatal period. Interestingly, westernisation was described by some participants as negatively impacting women's mental health, a new understanding not previously discussed that warrants further investigation.

This study found that traditional spiritual beliefs play a significant role in shaping how women understand and experience mental distress during the perinatal period, an area that has been neglected in previous health research. This underlines the need to consider the important intersection between spirituality and mental health. One potential way of achieving this is to ensure that any interventions or programmes are co-developed with the input of cultural experts who sit at the nexus between traditional health practices and the biomedical model of health. In the Gambian context, these includes experts such as Kanyeleng, who possess specialist knowledge of local spiritual beliefs, ritual, and musical practices associated with care for mothers and their infants.

## Limitations and future directions

Our approach to identifying idioms of distress was based on extensive consultation within the research team (which included two experienced mental health nurses) and with language experts at the NCAC. However, the identification and understanding of the idioms of distress deserves more extensive, in-depth research and analysis, as they shape the way that perinatal mental distress is experienced and understood. However, this was beyond the scope of the current project.

Some participants might have feared disclosing their own experiences of mental health within a focus group environment, especially as mental health problems are stigmatised in The Gambia. To help reassure participants, the facilitators instructed participants that information disclosed in the discussion should not be shared outside the group during the consenting process. Participants were also invited to talk about experiences they have heard from others rather than their own personal experiences.

Most FGDs were held at health centres and all pregnant participants were recruited from antenatal clinics. This may have led to some participants discussing treatments and concepts believed to be valued within the biomedical model of care. In addition, the sample of pregnant women was using the clinics. Therefore, we were not able to sample perceptions from women who were not seeking or could not seek care from the antenatal clinics in the first place. Future research could directly target women who do not use the clinics to better understand how their experiences might be similar or different to those discussed within this current work.

A culture-centred approach was employed in order to gain an understanding of Gambian women's experiences during the perinatal period that have been neglected in existing research. While much research on perinatal mental health in sub-Saharan Africa has represented local cultural traditions and practices [13, 49] as potential negative influences on a woman's mental state during and after pregnancy (e.g. gendered norms within households and practices such as polygamy [13, 49]), this research highlights the way in which women draw on local cultural traditions and practices as strategies for supporting mental health during the perinatal period. Moreover, these cultural practices, many of which involve music, provide insight into local understandings of the factors believed to contribute to mental distress during the perinatal period, including a breakdown in social relationships and spiritual problems caused by witchcraft or evil spirits. We argue that developing a fuller understanding of these local perspectives on experiences of perinatal mental distress, contributing factors, and coping strategies is necessary to inform culturally relevant global health initiatives that align with local priorities (e.g., [50, 51]). This is important in moving beyond outsider-driven responses that fail to adequately engage with local knowledge systems and culture. We argue that this is particularly necessary when addressing perinatal mental health, a neglected women's health issue subject to widespread stigma.

## Supporting information

**S1 Checklist. COREQ (COnsolidated criteria for REporting Qualitative research) checklist.**
(PDF)

**S1 Text. Focus group discussion guide by informant group.**
(DOCX)

**S2 Text. More detailed information about idioms of distress identified.**
(DOCX)

## Acknowledgments

We would like to thank the all the participants in the Gambia for sharing their knowledge and experiences with us. Thank you also to Hassoum Ceesay, Malick Gaye, Pa Bakary Sonko, Charlotte Hanlon, Paul Ramchandani, Ian Cross and Victoria Cornelius for their essential advice and support, and Jane Offerman for helping with the administration of this project.

## Author Contributions

**Conceptualization:** Katie Rose M. Sanfilippo, Bonnie McConnell, Vivette Glover, Lauren Stewart.

**Data curation:** Katie Rose M. Sanfilippo.

**Formal analysis:** Katie Rose M. Sanfilippo, Bonnie McConnell, Hajara B. Huma.

**Funding acquisition:** Lauren Stewart.

**Investigation:** Bonnie McConnell, Buba Darboe, Hajara B. Huma.

**Methodology:** Katie Rose M. Sanfilippo, Bonnie McConnell, Buba Darboe.

**Project administration:** Buba Darboe.

**Resources:** Katie Rose M. Sanfilippo, Buba Darboe, Lauren Stewart.

**Supervision:** Bonnie McConnell, Buba Darboe, Vivette Glover, Lauren Stewart.

**Writing – original draft:** Katie Rose M. Sanfilippo, Bonnie McConnell.

**Writing – review & editing:** Katie Rose M. Sanfilippo, Bonnie McConnell, Buba Darboe, Hajara B. Huma, Vivette Glover, Lauren Stewart.

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
