## [Decision Letter · Decision Letter 0]

3 May 2023

PGPH-D-22-01137

The experience of maternal mental distress in The Gambia: a qualitative study identifying terms of distress, perceptions of contributing factors and the supporting role of existing cultural practices

Dear Dr. Sanfilippo,

Thank you for submitting your manuscript to PLOS Global Public Health. After careful consideration, we feel that it has merit but does not fully meet PLOS Global Public Health’s publication criteria as it currently stands. Therefore, we invite you to submit a revised version of the manuscript that addresses the points raised during the review process.

Editor Comments:

This manuscript is very well-written and offers important insights into cultural manifestations of maternal mental health. The edits recommended by the reviewers will improve the clarity and potential for impact of the paper.I agree with the reviewer's suggestion to provide additional information on the participant observations in the methods section. It would also be helpful to provide the FGD guide(s) and final codebook as supplemental materials. See the COREQ guidelines for additional suggestions of information to add to the methods. Although the strengths of the approach are provided in the discussion, it should be paired with a description of the limitations of the study. Please also consider the recommendations related to the the Social Context theme.One of the reviewers provided additional detailed feedback in a pdf--please reach out if you cannot access or view this document.

We look forward to receiving your revised manuscript.

Kind regards,

Marie A. Brault, PhD

Academic Editor

Journal Requirements:

a. State what role the funders took in the study. If the funders had no role in your study, please state: “The funders had no role in study design, data collection and analysis, decision to publish, or preparation of the manuscript.”

b. If any authors received a salary from any of your funders, please state which authors and which funders.

Additional Editor Comments (if provided):

Reviewers' comments:

Reviewer's Responses to Questions

**Comments to the Author**

1. Does this manuscript meet PLOS Global Public Health’s publication criteria? Is the manuscript technically sound, and do the data support the conclusions? The manuscript must describe methodologically and ethically rigorous research with conclusions that are appropriately drawn based on the data presented.

Reviewer #1: Yes

Reviewer #2: Yes

Reviewer #3: Yes

2. Has the statistical analysis been performed appropriately and rigorously?

Reviewer #1: N/A

Reviewer #2: Yes

Reviewer #3: N/A

3. Have the authors made all data underlying the findings in their manuscript fully available (please refer to the Data Availability Statement at the start of the manuscript PDF file)?

Reviewer #1: Yes

Reviewer #2: Yes

Reviewer #3: Yes

4. Is the manuscript presented in an intelligible fashion and written in standard English?

Reviewer #1: Yes

Reviewer #2: Yes

Reviewer #3: Yes

5. Review Comments to the Author

Reviewer #1: This is a well-written paper which explores local conceptualisations of perinatal distress, as well as the economic and sociocultural factors which can impact on perinatal mental health and wellbeing. The study is well-conducted and will be an important contribution to the field. Below are some comments/suggestions, which should be considered prior to publication.

• Lines 83-87 it is stated that “Moreover, specific social factors in various LMICs present additional risks to maternal mental health. For example, a study by Wittkowski et al. (15) identified four main factors contributing to postnatal depression symptoms in Sub-Saharan Africa: lack of social support (especially from the husband), relationship problems, an unwanted pregnancy, and cultural factors, such as family structures or polygamy.” – it might be argued that (apart from polygamy) that these are universal predictors of poor maternal mental health, not just specific to LMICs; therefore perhaps be more tentative with this statement.

• Line 126: I recommend that “griot” is defined briefly here for those not familiar with the term

• Lines 177-178: it is noted that the research team engaged in participant observation to gain an understanding of cultural ceremonies in the perinatal period. Are these observations included as part of the analysis in the current paper? If so, more information about these observations should be included (e.g. which ceremonies, frequency etc).

• Line 188 – how are research sites defined (e.g. geographical location/setting?)

• There is no information about how the different participants were recruited/identified, and the procedures involved. I recommend that the authors looks at guidelines for the reporting if qualitative studies (e.g. COnsolidated criteria for REporting Qualitative research (COREQ) checklist; Tong et al., 2007) to ensure important information is reported in the manuscript (e.g. relationship of interviewees to participants etc). A copy of the topic guide should also be included (i.e. as supplementary file)

• There was very little information regarding the demographic characteristics of participants, especially the pregnant women (e.g. age, parity, marital status, employment), which are likely to have an impact on their experiences of pregnancy and motherhood. If these have been collected, it would be helpful if these can be summarised in the manuscript. Similarly, with other participants it might be helpful to know some of their demographic (e.g. age/gender) and professional (where appropriate) characteristics (e.g. how long have they been a midwife).

• Lines 239-241: were there any differences in the conceptualisations of distress across the different FGDs or was this the general consensus? I think the explanation around stigma and the different idioms used to describe distress could be expanded on – did these perceptions of stigma come from the different groups, or from the research team?

• I am not convinced that “Social Context” necessitates a distinct theme from the other themes i.e. Economic Factors, Social Factors, and Spiritual Factors, as the wider social context intersects with, and is important within, these three themes (e.g. pregnancy out of wedlock is discussed in social context but is also a social factor). Please consider whether a separate theme is needed here, or if the theme needs further refinement/development (e.g. healthcare systems seems important and distinct from the others)

• Lines 324-327: it is not clear how these cultural shifts in values can negatively (or positively) impact women and/or mothers – can this argument be developed further?

• I would have expected to see a discussion/recognition of the strengths and limitations of the research. How were quality and trustworthiness ensured in this study?

• What implications for future research does this project have? What are the recommendations for researchers working in this field?

• Ensure statements in the discussion are grounded in the data. For example, “hiding pregnancies” (Line 659), is not described in the results.

Reviewer #2: Title: The experience of maternal mental distress in The Gambia: a qualitative study identifying

terms of distress, perceptions of contributing factors and the supporting role of existing cultural

practices

I find this a very interesting and relevant study that is very worth publishing, but I have some comments and suggestions that I have made on the PDF itself which would contribute towards increased scientific rigour and reliability of your reporting and of the article itself. Once all of these comments have been addressed I would be very happy to see this paper published. I do not necessarily need to see your responses to these comments.

Page 7:

I would be interested to know how you got to identifying the idioms of distress when it doesn’t appear to have been a particular theme or line of questioning in your focus group interview schedules (unless I missed it).

Page 8: you make no mention of any results from your participant observation methods – please either incorporate this throughout or remove this as a method.

Page 11: results:

I have a query about the structure of your results – I have not come across the way you have presented the results before in mental health literature – i.e. incorporating external data and literature in your results. This may be acceptable by the journal but I would then tend towards labelling this section as ‘results and discussion’ rather than just results, and then move some of your discussion up into the current ‘result’s section, and then add a concluding section to sum up your findings and recommendations for future research, practice, and policy.

Page 24

I have made a few substantial comments about the theme ‘cultural practices’ and its singular orientation towards music.

Discussion and conclusion

I would also like to see any limitations of the study mentioned. (e.g. the fact that the data is quite a few years old)

Reviewer #3: Thank you for sending this paper - I thoroughly enjoyed reading it, and recommend for publication. This is a beautifully researched and written paper which highlights very clearly a vital issue in maternal mental health, the importance of understanding the cultural context.

I struggle to find issues to revise for review, to be honest, and can recommend this paper as it is for publication. The following are my reflection on this article, rather than issues for correction;

For myself I would have been interested in further elaboration of the linguistic issues, the idioms used and why each term was more or less stigmatising.

I would also have been interested in further description, and perhaps a stronger stress upon, the power relations inherent in the family relationships within homes/compounds. The difficulty for (especially young) women in being given permission to attend medical appointments is something I remember well, and saw starkly for those who had tested HIV+ in pregnancy. For many women and girls to manage to attend appointments takes a persistence and sometimes bravery that should not be understated.

While I agree and appreciate that music is key to understanding the ways in which cultural values are transmitted, and support and community offered to women through these, I also felt that the importance of food in these ceremonies and in the cohesion of social life could also come through more strongly. There are important and insightful mentions throughout of the ways in which food plays into these stresses and supports, and these could be tied together in the discussion, in a similar way to the theme of music and culture. Sitting together and eating together are demonstrations of community - and being alone or isolated is more than just stressful. In my research people talked about the isolation of HIV stigma - and specifically eating alone - being as deadly as the disease itself.

Similarly the performative nature of clothing and ceremonies - being able to host a ceremony and have new clothes, could be made clearer - although this sits within the economic factors section there are broader societal norms at play here as well, that may not be clear to many readers.

I also wondered about the degree to which this project was able to make comment on those women and girls who were more or less able or likely to access care - I appreciate this may be out of scope for this research, but would be interesting to understand further.

6. PLOS authors have the option to publish the peer review history of their article (what does this mean?). If published, this will include your full peer review and any attached files.

**Do you want your identity to be public for this peer review?** For information about this choice, including consent withdrawal, please see our Privacy Policy.

Reviewer #1: No

Reviewer #2: No

Reviewer #3: **Yes: **Rebecca Cassidy

---

## [Decision Letter · Decision Letter 1]

7 Aug 2023

The experience of maternal mental distress in The Gambia: a qualitative study identifying idioms of distress, perceptions of contributing factors and the supporting role of existing cultural practices

PGPH-D-22-01137R1

Dear Dr Sanfilippo,

We are pleased to inform you that your manuscript 'The experience of maternal mental distress in The Gambia: a qualitative study identifying idioms of distress, perceptions of contributing factors and the supporting role of existing cultural practices' has been provisionally accepted for publication in PLOS Global Public Health.

Best regards,

Marie A. Brault, PhD

Academic Editor

Reviewer Comments (if any, and for reference):

Reviewer's Responses to Questions

**Comments to the Author**

1. If the authors have adequately addressed your comments raised in a previous round of review and you feel that this manuscript is now acceptable for publication, you may indicate that here to bypass the “Comments to the Author” section, enter your conflict of interest statement in the “Confidential to Editor” section, and submit your "Accept" recommendation.

Reviewer #1: All comments have been addressed

2. Does this manuscript meet PLOS Global Public Health’s publication criteria? Is the manuscript technically sound, and do the data support the conclusions? The manuscript must describe methodologically and ethically rigorous research with conclusions that are appropriately drawn based on the data presented.

Reviewer #1: Yes

3. Has the statistical analysis been performed appropriately and rigorously?

Reviewer #1: N/A

4. Have the authors made all data underlying the findings in their manuscript fully available (please refer to the Data Availability Statement at the start of the manuscript PDF file)?

Reviewer #1: Yes

5. Is the manuscript presented in an intelligible fashion and written in standard English?

Reviewer #1: Yes

6. Review Comments to the Author

Reviewer #1: The authors have addressed comprehensively all the comments provided by the editor and the three reviewers.

7. PLOS authors have the option to publish the peer review history of their article (what does this mean?). If published, this will include your full peer review and any attached files.

**Do you want your identity to be public for this peer review?** For information about this choice, including consent withdrawal, please see our Privacy Policy.

Reviewer #1: No
